# The Short-Term Impact of Neoadjuvant Chemotherapy on the Outcome of Patients Undergoing Pneumonectomy for Lung Cancer: Is It Acceptable Nowadays?

**DOI:** 10.3390/jcm14072419

**Published:** 2025-04-01

**Authors:** Antonio Mazzella, Sebastiano Maiorca, Giuseppe Nicolosi, Patrick Maisonneuve, Antonio Passaro, Monica Casiraghi, Luca Bertolaccini, Filippo de Marinis, Lorenzo Spaggiari

**Affiliations:** 1Division of Thoracic Surgery, IEO, European Institute of Oncology IRCCS, 20141 Milan, Italy; sebastiano.maiorca@unimi.it (S.M.); giuseppe.nicolosi@ieo.it (G.N.); monica.casiraghi@ieo.it (M.C.); luca.bertolaccini@gmail.com (L.B.); lorenzo.spaggiari@ieo.it (L.S.); 2Department of Oncology and Haemato-Oncology, University of Milan, 20122 Milan, Italy; patrick.maisonneuve@ieo.it; 3Division of Thoracic Oncology, IEO, European Institute of Oncology IRCCS, 20141 Milan, Italy; antonio.passaro@ieo.it (A.P.); filippo.demarinis@ieo.it (F.d.M.); 4Division of Epidemiology and Biostatistics, IEO, European Institute of Oncology IRCCS, 20141 Milan, Italy

**Keywords:** lung cancer, lung surgery, pneumonectomy, neoadjuvant treatment, chemotherapy

## Abstract

**Objective:** We aimed at assessing our experience at the European Institute of Oncology in order to evaluate the peri- and immediately post-operative impact of neoadjuvant chemotherapy in patients who underwent pneumonectomy for NSCLC. **Materials and methods:** We retrospectively reviewed the outcomes and medical records of patients undergoing pneumonectomy (2010–2024). We compared pre-, peri- and post-operative outcomes of patients treated with induction chemotherapy and subsequent pneumonectomy with patients who underwent surgery directly. Differences in their distribution between study arms were assessed using the chi-square test for categorical variables or the Mantel–Haenszel test for trend for ordinal variables. We tested normality of the distribution of continuous variables using the Shapiro–Wilk test. We used logistic regression to quantify the risk of various outcomes (complications, 30-day and 12-day mortality) in patients who received neoadjuvant chemotherapy. Risks were expressed as odds ratios (ORs) with 95% confidence intervals (CIs adjusted for age (<60, 60–64, 65–69, ≥70 years), sex and comorbidities (cardiovascular, pulmonary or previous cancer). **Results:** We observed a higher frequency of post-operative respiratory complications in patients who underwent neoadjuvant therapy and pneumonectomy compared to those who only underwent surgery (11.4% vs. 18.5%; *p* = 0.05). After adjustment for age, sex and comorbidities we observed a significantly higher rate of pulmonary complications (OR 1.95; 95% CI 1.09–3.47; *p* = 0.02), ARDS (OR 2.88; 95% CI 1.26–6.59; *p* = 0.02) and 30-day mortality rate (OR 8.19; 95% CI 1.33–50.3; *p* = 0.02) in pre-treated patients. **Conclusions:** It is therefore strongly recommended to study and select potentially eligible patients in an extremely meticulous way before starting the neoadjuvant treatment, and to thoroughly re-evaluate the cardiorespiratory status after inductive therapy, before surgery.

## 1. Introduction

Despite the notable improvement in cancer screening, diagnostic and surgical techniques and innovative therapies, advanced-stage lung cancer still represents one of the major challenges in achieving a cure.

In this segment of patients, integrated treatments are essential, and neoadjuvant therapy is often considered as forerunner in the therapeutic path. In the case of advanced neoplasms, large in size and infiltrating nearby structures (T3–T4), which are often associated with metastatic lymph nodes (N2), pre- and peri-operative treatments are crucial to reduce tumor volume, enhance resectability, and improve disease-free and overall survival rates [1,2].

However, despite the downstaging linked to the neoadjuvant treatment, in some cases pneumonectomy remains the only surgical therapeutic alternative to obtain oncologic radicality. This intervention is unanimously recognized as a high-risk operation, in terms of peri- and post-operative morbidity [3,4] and in terms of mortality (5–9% among all pulmonary resections) [5,6,7,8,9].

It is equally recognized that induction chemotherapy, usually consisting of platinum doublet with vinorelbine/pemetrexed, may have an important clinical impact on the general status of pneumonectomised patients, playing a key role in the onset of post-operative infectious complications [3,6] or broncho–pleural fistula onset [8].

Similarly, the role of neoadjuvant treatment in the development of post-pneumonectomy respiratory complications is still controversial [2,3,4,7,8,10].

In the modern era, along with the advent of new drugs, with the implementation of new immunotherapy protocols, we are witnessing a Copernican revolution. However, even if chemotherapy is now associated with immunotherapy [11,12,13,14,15] in neoadjuvant setting, its possible harmful effects on pneumonectomised patients remain to be investigated.

In this view, we aimed at assessing our practice at the IEO (European Institute of Oncology) for assessing the peri- and immediately post-operative impact of neoadjuvant chemotherapy in patients undergoing pneumonectomy for NSCLC.

## 2. Materials and Methods

### 2.1. Patients

This is a retrospective study, according to the STROBE (Strengthening the Reporting of Observational studies in Epidemiology) statement, regarding our single-center experience [16]. From our database, we retrospectively revised pre-, peri- and immediate post-operative outcomes and medical and operative records (Table 1 and Table 2). Written informed consent to use clinical data for scientific purposes were obtained from all patients before surgery. The Ethics Committee of the European Institute of Oncology approved the study (UID 4436).

A total of 406 patients were treated through pneumonectomy for lung cancer between 2010 and 2024 at the European Institute of Oncology. Pre-operative staging consisted of a total body computed tomography scan (CT-scan), positron emission tomography (PET) with fluorodeoxyglucose (FDG) and cardiological and pulmonary evaluation. We added perfusional pulmonary scintigraphy and a cardio-pulmonary exercising test (CPET) as routine exams, starting in 2017.

During multidisciplinary meetings, thoracic surgeons and oncologists confirmed patients’ operability and/or medical treatment.

We performed pneumonectomy, via lateral thoracotomy and a radical mediastinal lymph node dissection in all patients (station 4R, 7, 8, 9, 10 on the right; 5, 6, 7, 8, 9, 10 on the left).

After surgery, patients were immediately extubated in the operating or recovery room, and then transferred to the thoracic department or to the intensive care unit (ICU). If for respiratory/anesthesiology problems, the extubation was not achieved in the OR, the double-lumen tube was replaced by single-lumen tube (tidal volume 8 mL/kg, PEEP 5 cm H_2_O), and the patient was moved to the ICU. The pleural cavity was routinely drained.

### 2.2. Immediate Post-Operative Course

The peri-operative management was standardized: early mobilization and deambulation, thrombotic prophylaxis, chest physiotherapy and adequate pain control. The chest drain was removed on day 7 after surgery, after excluding suspect bronchopleural fistula by bronchoscopy; the patient was discharged the day after, unless contraindications occurred. All patients were reevaluated 3 and 6 months after discharge on an outpatient basis.

### 2.3. Morbidity and Mortality

Morbidity was defined as occurrence of grade II or more Clavien–Dindo [17] pulmonary complications within 30 days from surgery. Concerning respiratory complications included the following:-Acute respiratory failure, defined as post-operative necessity of non-invasive ventilation or more than 12 h of mechanically assisted ventilation;-Pneumonia (presence of three of the following criteria: pulmonary thickening on chest X-ray, hyperpyrexia > 38 °C, leukocytosis, purulent secretions, bacterial isolation from the sputum or bronchoaspirate);-Lung atelectasis requiring bronchoscopy;-Pulmonary edema;-Pulmonary embolism.

Acute respiratory distress syndrome (ARDS) and acute lung injury (ALI) are defined as respiratory failure with acute onset, arterial oxygen tension/fraction of inspired oxygen less than 200 mm Hg (for ARDS) or 300 mm Hg (for ALI) and are associated with radiological findings of pulmonary infiltrates.

Other complications considered were cardiac (arrhythmia, acute myocardial ischemia, cardiac failure needing for inotropic support), surgical (bronchial fistula) or other. Mortality was defined as in-hospital death or death within 30 days from surgery. The 120-day mortality was defined as death within 120 days from surgery, after being discharged from the hospital.

### 2.4. Indications for Pre-Operative Management/Treatment

Pre-operative therapies in our institution were decided on an individual-based discussion during a weekly Multidisciplinary Tumor Board.

We usually use a platinum/gemcitabine or platinum/pemetrexed doublet as standard therapy for all patients (3 or 4 cycles before surgery). As already debated in other papers or in other international guidelines [1,2], neoadjuvant chemotherapy administered before surgical intervention has been shown to reduce tumor volume, enhance resectability and improve disease-free and overall survival rates [2], and it is usually reserved for locally advanced stages (IIIA stage, N2 lymph node involvement) at the time of diagnosis that are potentially resectable. Indeed, one of the main objectives of neoadjuvant chemotherapy is to achieve disease downstaging, in terms of size and extension to nearby organs, in order to reduce the extent of the subsequent surgical resection. This approach also enables an early evaluation of tumor sensitivity to systemic treatments, providing valuable prognostic insights for post-operative follow-up.

### 2.5. Patient Follow-Up

Patients underwent clinical examination and a chest X-ray was performed after one month. Follow-up included physical examination, tumor blood markers and a computed tomography scan (every 3 months in the first year, every 6 months in the following years).

### 2.6. Statistical Analysis

Participants’ characteristics and outcomes were presented as numbers and frequencies and differences in their distribution between study arms were assessed using the chi-square test for categorical variables or the Mantel–Haenszel test for trend for ordinal variables. We tested normality of the distribution of continuous variables using the Shapiro–Wilk test. As distributions were not normally distributed, we presented median values with interquartile range and tested differences across groups using the non-parametric sign-rank test for medians.

We used logistic regression to quantify the risk of various outcomes (complications, 30-day and 12-day mortality) in patients who received neoadjuvant chemotherapy. Risks were expressed as odds ratios (ORs) with 95% confidence intervals (CIs adjusted for age (<60, 60–64, 65–69, ≥70 years), sex and comorbidities (cardiovascular, pulmonary or previous cancer).

Analyses were performed with the SAS software version 9.4 (Cary, NC, USA). All tests were two sided and *p* < 0.05 was considered to be statistically significant.

## 3. Results

We performed 405 pneumonectomies in the examined period (202 right and 204 left pneumonectomies). Post-operative pathological results were 147 NSCLC, 127 adenocarcinomas (64%), 108 squamous cell carcinomas (33.3%), 4 adenosquamous carcinomas (1.6%), and 20 undifferentiated NSCLC tumors (1.1%). Tracheal sleeve pneumonectomy was performed in 13 cases. All clinical and demographic data are expressed in Table 1.

A total of 195 patients out of 406 (48%) received neo-adjuvant treatment. The 211 remaining patients (52%) underwent upfront pneumonectomy.

There are no significant differences between the two cohorts regarding pre-operative and peri-operative data, expect for age, which was significantly higher in the no-neoadjuvant treatment cohort (*p* = 0.03).

Post-operative comorbidity and 30- and 120-day mortality are reported in Table 2. Particularly, 30-day mortality of the whole cohort was 2.2% (9 patients); 120-day mortality was 7.6% (31 patients).

We observed a higher frequency of post-operative respiratory complications in patients who underwent neoadjuvant therapy and pneumonectomy, compared to those who only underwent surgery (11.4% vs. 18.5% *p*: 0.05). A similar trend, although not reaching statistical significance, was observed for 30-day mortality (1% vs. 3.6% *p*: 0.09) and the development of post-operative ARDS (4.8 vs. 9.7% *p* = 0.06).

Considering the adjusted analysis (age, sex, comorbidities), we observed a significantly higher rate of pulmonary complications (OR 1.95; 95% CI 1.09–3.47; *p* = 0.02), ARDS (OR 2.88; 95% CI 1.26–6.59; *p* = 0.02) and 30-day mortality rate (OR 8.19; 95% CI 1.33–50.3; *p* = 0.02) in pre-treated patients.

There is no impact of neoadjuvant CT in the 120-day mortality (7.1% vs. 8.2%; *p* = 0.71).

## 4. Discussion

The role of induction therapy as a risk factor predisposing the development of respiratory complications in patients undergoing lung resection is still source of debate and controversy. This debate is even more divisive when induction therapy is followed by a pneumonectomy. Indeed, this issue becomes critical in patients undergoing pneumonectomy, considering that any event that negatively affects lung function could be life threatening [11,18].

Some previous studies demonstrated that the incidence of cardiopulmonary complications (e.g., ARDS) was similar [1,3,8,19,20,21,22] or slightly higher [12] in pre-treated patients compared to those without neoadjuvant therapy (21% vs. 18%).

On the other hand, in other series, neoadjuvant therapy was significantly associated with the development of pulmonary complications, which were even three times more frequent in patients who received induction chemotherapy than in those who did not [4,11,23].

Even in one of the largest and best meta-analyses about this topic [2], authors discuss in detail the role of neoadjuvant therapy and post-operative mortality, but there is no consensus about its effective role, as a risk factor, for the onset of respiratory complications. The only conclusion was that pulmonary complications were the most common cause of death at 30 and 90 days.

The spectrum of possible chemotherapy-related lung damages is obviously broad, ranging from subclinical modifications to an acute toxicity (very rare in patients with lung cancer, <1%), which are more frequent in older and smoker patients or patients with pre-existing interstitial lung disease [24,25].

Referring to the subclinical modifications mentioned above, the most frequent lung damage is at the level of the alveolus–capillary membrane (80%). This condition is almost always asymptomatic, and the only way to detect it is by diffusion of carbon monoxide assessment (DLCO) [2,26]. It has been demonstrated that induction therapy causes DLCO reduction in lung cancer in the order of 15% but sometimes as high as 40% to 50%, both in lung cancer [11,27,28,29] as in other malignancies [30].

If on the one hand these changes remain silent and subclinical, on the other hand they increase the risk of pulmonary morbidity in the case of sequential treatments (e.g., radiotherapy or surgery) [31].

From a molecular point of view, chemotherapy promotes the presence of inflammatory mediators and cells, such as interleukins or cytotoxic factors, at the level of the alveolus–capillary membrane, facilitating the entrance of infective agents or promoting intra-alveolar exudates [25,26]. On the other hand, it also determines an increased systemic inflammatory response [32]. Starting from this inflammatory environment, the subsequent oxidative stress could trigger an acute, subacute or chronic response, depending on the degree of injury, pre-existing conditions and capacity of repair. It is therefore easy to understand that further stress, such as local therapies (radiotherapy or surgery) can fuel this condition, especially in patients affected by chronic lung injury [10,30]. Starting from this inflammatory assumption, the right counterbalance would be the use of anti-inflammatory drugs (corticosteroids), yet it is well known how they could have a detrimental effect in pneumonectomised patients (adverse effect on bronchial healing and bleeding).

This subclinical process impacts the histological and anatomopathological composition of the lung tissue. It has demonstrated that chemotherapy can cause severe lung injury, predisposing patterns of severe and diffuse interstitial involvement, interstitial chronic inflammation, fibrosis and emphysematous changes, foci of bronchiolitis obliterans-organizing pneumonia (BOOP), diffuse alveolar damage (DAD), desquamative interstitial pneumonia (DIP)-like reactions or usual interstitial pneumonia (UIP)-like changes [25].

In light of the possible new respiratory scenario arising after inductive therapy, an “ex novo” deep cardio-respiratory study (spirometry, DLCO, CPET and lung perfusion scan) is essential in view of a subsequent pneumonectomy. In particular, we perform a complete post-induction therapy reassessment, consisting of a CPET and lung perfusion scan. This is essential to evaluate the pre-operative (cut-off VO2 max > 15 mL/min/kg) and predicted-post-operative (cut-off VO2max > 10 mL/min/kg) VO2max, more than DLCO and simple spirometry, in order to evaluate respiratory post-operative morbidity and mortality. Indeed, the CPET provides a holistic assessment of the patient’s physiological status, depending on several factors: respiratory, cardiovascular, musculoskeletal, circulatory, training and effort.

In our analysis the correlation between induction therapy and incidence of postoperative respiratory complications is very clear, both in the simple and adjusted analysis. In particular, the risk of developing simple respiratory complications is double. Concerning ARDS, but it becomes even triple, even though statistical significance was not reached in this last subgroup of patients. Indeed, we observed a trend of significance that suggests to us to understand how the development of ARDS can be strictly linked to pre-operative chemotherapy. This means that a pre-treated patient will have a three times greater risk of developing post pneumonectomy ARDS.

Regarding 30-day mortality, even if there is a strong significance, both in univariate (*p* = 0.09) and in adjusted analysis (*p* = 0.02), the number of the events is too small (nine) for stating a certain association between induction therapy and mortality. However, in the majority of cases (7 out of 9 cases), the main cause of death is linked to a respiratory complication (ARDS).

There are no doubts on the role of the inductive chemotherapy, alone or in association with other treatments, in giving an advantage in terms of survival of this particular cohort of patients [1,2,12,13,14,15,19,32] compared to a perhaps lower advantage in the adjuvant setting, in patients undergoing pneumonectomy [10,20].

However, we are now facing a new era in the medical and surgical treatment of locally advanced lung cancer. If previously the standard was the classical neoadjuvant chemotherapy and surgery, with the advent of immunotherapy in the last decade we are witnessing a real Copernican revolution. The combination of pre- or peri-operative immunotherapy and chemotherapy has become dominant in recent years [13,14,15,16], with surprising results compared to standard chemotherapy in terms of efficacy and survival. However, this combination made the subsequent surgery, “de facto”, much more complex and difficult due to the extensive subsequent fibrotic or inflammatory reactions, with an acceptable postoperative complications rate, except that the pneumonectomy was identified as an important risk factor [15].

Nevertheless, there are other questions to be asked: what is the price of this advantage in patients undergoing pneumonectomy? Are all patients considered suitable for surgery before induction therapy? Is it also considered suitable afterwards? Indeed, in this optic, the need for pneumonectomy (based on pre-operative staging) is sometimes viewed as a contraindication to performing neo-adjuvant chemo-immunotherapy.

We must then consider patients with diseases sensitive to targeted therapies (EGFR, ALK), in which the prognosis could be linked to the molecular therapy rather than to the intervention. From this perspective, therefore, neoadjuvant chemotherapy, alone, with all its burden of complications, would lose even more meaning.

From this perspective, it is therefore mandatory to carefully select potentially pneumonectomy eligible patients before the start of the neoadjuvant chemotherapy in order to minimize the peri-operative and post-operative risks of a surgical procedure that is already highly invasive and dangerous.

In light of the strong association between 30-day mortality and respiratory morbidity, chemotherapy-related in this particular cohort of patients, it will be necessary in the future to investigate even more on precision medicine, looking for additional molecular targets (beyond those already known). The final goal will be able to perform an induction therapy no longer based on chemotherapy, but rather on target drugs or on immunotherapy alone, perhaps relegating chemotherapy to a second line therapy, given the toxicity it presents. It is very clear that many other studies are needed to support these hypotheses, which now might seem almost blasphemous.

There are clearly some limitations to this study: firstly, its retrospective design. In addition, we did not include an extensive pre-operative pulmonary functionality in the analysis, because the CPET and pulmonary scintigraphy were routinely executed only since 2017. However, this represents a uniform cohort of patients, whose intra- and post-operative management was similar, adapted to the protocols of our department.

## 5. Conclusions

Pneumonectomy remains a safe procedure. However, in cases of induction chemotherapy, the rate of associated respiratory complications increases, as well as the 30-day mortality. It is therefore strongly recommended to study and select potentially eligible patients in an extremely meticulous way before starting the neoadjuvant treatment, and to thoroughly re-evaluate the cardiorespiratory status after inductive therapy, before surgery.

## Figures and Tables

**Table 1 jcm-14-02419-t001:** Characteristics of NSCLC patients who received pneumonectomy for primary lung cancer during 2010–2024.

	Total	Neoadjuvant Treatment		*p* Value
		No	Yes	
All	406 (100.0)	211 (100.0)	195 (100.0)	
Year of surgery				
2010–2014	183 (45.1)	86 (40.8)	97 (49.7)	
2015–2024	223 (54.9)	125 (59.2)	98 (50.3)	0.07
Age group				
<60	112 (27.6)	53 (25.1)	59 (30.3)	
60–64	72 (17.7)	32 (15.2)	40 (20.5)	
65–69	110 (27.1)	55 (26.1)	55 (28.2)	
70+	112 (27.6)	71 (33.6)	41 (21.0)	0.03
Sex				
Male	301 (74.1)	164 (77.7)	137 (70.3)	
Female	105 (25.9)	47 (22.3)	58 (29.7)	0.09
Cardiovascular comorbidity				
No	269 (66.3)	131 (62.1)	138 (70.8)	
Yes	134 (33.0)	79 (37.4)	55 (28.2)	0.05
Unknown	3 (0.7)	1 (0.5)	2 (1.0)	
Pulmonary comorbidity				
No	362 (89.2)	188 (89.1)	174 (89.2)	
Yes	41 (10.1)	22 (10.4)	19 (9.7)	0.83
Unknown	3 (0.7)	1 (0.5)	2 (1.0)	
Previous cancer				
No	346 (85.2)	175 (82.9)	171 (87.7)	
Yes	57 (14.0)	35 (16.6)	22 (11.3)	0.13
Unknown	3 (0.7)	1 (0.5)	2 (1.0)	
Histology				
NSCLC	147 (36.2)	78 (37.0)	69 (35.4)	
Squamous	108 (26.6)	58 (27.5)	50 (25.6)	
Adenocarcinoma	127 (31.3)	61 (28.9)	66 (33.8)	
Adenosquamous	4 (1.0)	-	4 (2.1)	
Other NSCLC	20 (4.9)	14 (6.6)	6 (3.1)	0.09
Stage				
Stage I	23 (5.7)	9 (4.3)	14 (7.2)	
Stage II	101 (24.9)	59 (28.0)	42 (21.5)	
Stage III	271 (66.7)	141 (66.8)	130 (66.7)	Trend
Stage IV	1 (0.2)	-	1 (0.5)	0.99
Missing	10 (2.5)	2 (0.9)	8 (4.1)	
Side				
Right	202 (49.8)	98 (46.4)	104 (53.3)	
Left	204 (50.2)	113 (53.6)	91 (46.7)	0.17
Extensive resection				
No	320 (78.8)	173 (82.0)	147 (75.4)	
Yes	86 (21.2)	38 (18.0)	48 (24.6)	0.10
Sleeve				
No	392 (96.6)	207 (98.1)	185 (94.9)	
Yes	13 (3.2)	4 (1.9)	9 (4.6)	0.12
Unknown	1 (0.2)	0 (0.0)	1 (0.5)	
ICU				
None	174 (42.9)	88 (41.7)	86 (44.1)	
1 day	195 (48.0)	106 (50.2)	89 (45.6)	
2–5 days	20 (4.9)	10 (4.7)	10 (5.1)	Trend
>5 days	12 (3.0)	4 (1.9)	8 (4.1)	0.40
Unknown	5 (1.2)	3 (1.4)	2 (1.0)	
FEV1, mean ± SD	2.33 ± 0.69	2.30 ± 0.73	2.36 ± 0.64	
FEV1, median (IQR)	2.24 (1.87–2.75)	2.17 (1.77–2.71)	2.27 (1.97–2.77)	0.23
FEV1%, mean ± SD	82.5 ± 18.9	82.4 ± 18.9	82.5 ± 19.0	
FEV1%, median (IQR)	83.0 (68.8–97.0)	81.2 (68.0–97.0)	85.8 (70.0–97.0)	0.22

**Table 2 jcm-14-02419-t002:** Complications and 30- and 120-day mortality of patients undergoing pneumonectomy, with or without neoadjuvant treatment.

	Total	Neoadjuvant Treatment		*p* Value
		No	Yes	
All	405 (100)	210 (100)	195 (100)	
Complications				
None	212 (52.4)	116 (55.2)	96 (49.2)	
Any	193 (47.6)	94 (44.8)	99 (50.8)	0.23
Adj RR (95% CI) *		1.00	1.38 (0.92–2.08)	0.12
Respiratory				
No	345 (85.2)	186 (88.6)	159 (81.5)	
Yes	60 (14.8)	24 (11.4)	36 (18.5)	0.05
Adj RR (95% CI) *		1.00	1.95 (1.09–3.47)	0.02
Cardiac				
No	325 (80.0)	172 (81.5)	153 (78.5)	
Yes	81 (20.0)	39 (18.5)	42 (21.5)	0.44
Adj RR (95% CI) *		1.00	1.46 (0.87–2.44)	0.15
Other				
No	304 (72.3)	158 (75.2)	146 (75.3)	
Yes	100 (24.7)	52 (24.8)	48 (24.7)	1.00
Adj RR (95% CI) *		1.00	1.07 (0.66–1.72)	0.79
Fistula				
No	360 (88.9)	188 (89.5)	172 (88.2)	
Yes	45 (11.1)	22 (10.5)	23 (11.8)	0.75
Adj RR (95% CI) *		1.00	1.34 (0.70–2.57)	0.37
ARDS				
No	376 (92.8)	200 (95.2)	176 (90.3)	
Yes	29 (7.2)	10 (4.8)	19 (9.7)	0.06
Adj RR (95% CI) *		1.00	2.88 (1.26–6.59)	0.01
30-day mortality				
No	397 (97.8)	209 (99.0)	188 (96.4)	
Yes	9 (2.2)	2 (1.0)	7 (3.6)	0.09
Adj RR (95% CI) *		1.00	8.19 (1.33–50.3)	0.02
120-day mortality				
No	375 (92.4)	196 (92.9)	179 (91.8)	
Yes	31 (7.6)	15 (7.1)	16 (8.2)	0.71
Adj RR (95% CI) *		1.00	1.30 (0.59–2.84)	0.52

* Adjusted for age (<60, 60–64, 65–69, 70+), sex and comorbidities (cardiovascular, pulmonary or previous cancer).

## Data Availability

No new data was created.

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
