# Peer review of "The Short-Term Impact of Neoadjuvant Chemotherapy on the Outcome of Patients Undergoing Pneumonectomy for Lung Cancer: Is It Acceptable Nowadays?"

_jcm, 2025, doi:10.3390/jcm14072419_

Round 1

Reviewer 1 Report

Comments and Suggestions for Authors

Antonio Mazzella and team has done a great job. Manuscript is well designed and well written. This article will provide a new perspective for clinicians while making a decision for neoadjuvant chemotherapy in NSCLC patents.

Author Response

REVIEWER 1

Antonio Mazzella and team has done a great job. Manuscript is well designed and well written. This article will provide a new perspective for clinicians while making a decision for neoadjuvant chemotherapy in NSCLC patents.

Thanks for your remark and appreciation.

Reviewer 2 Report

Comments and Suggestions for Authors

Comments and Suggestions for the Authors

  1. Clarification of the Impact of Neoadjuvant Chemotherapy on Pneumonectomy Outcomes
    The study provides valuable insights into the relationship between neoadjuvant chemotherapy and postoperative complications in patients undergoing pneumonectomy. However, the analysis would benefit from a more detailed stratification of chemotherapy regimens and their differential impact on clinical outcomes. Including data on whether certain regimens are more strongly associated with postoperative complications would enhance the study’s applicability and provide a more nuanced understanding of the risks involved.

  2. Contextualizing the Study within Current NSCLC Treatment Strategies
    Given the significant advancements in NSCLC treatment, particularly with the introduction of immunotherapy, the discussion would benefit from an assessment of its role in the neoadjuvant setting. The study could compare the impact of neoadjuvant chemotherapy alone versus combination therapies, particularly in relation to postoperative complications and survival outcomes. Expanding this aspect would strengthen the study’s clinical relevance and place the findings within the context of modern treatment paradigms.

Author Response

Clarification of the Impact of Neoadjuvant Chemotherapy on Pneumonectomy Outcomes

The study provides valuable insights into the relationship between neoadjuvant chemotherapy and postoperative complications in patients undergoing pneumonectomy. However, the analysis would benefit from a more detailed stratification of chemotherapy regimens and their differential impact on clinical outcomes. Including data on whether certain regimens are more strongly associated with postoperative complications would enhance the study’s applicability and provide a more nuanced understanding of the risks involved.

Thanks for the important remark. We usually use a platinum/gemcitabine or platinum/pemetrexed doublet and this is our standard therapy for all patients (3 or 4 cycles before surgery). We do not sub differentiate gemcitabine/pemetrexed groups (very few patients underwent other CT treatment i.e. paclitaxel) because we did not found any difference.

However, we add some specifics in the material and methods paragraph.

Contextualizing the Study within Current NSCLC Treatment Strategies

Given the significant advancements in NSCLC treatment, particularly with the introduction of immunotherapy, the discussion would benefit from an assessment of its role in the neoadjuvant setting. The study could compare the impact of neoadjuvant chemotherapy alone versus combination therapies, particularly in relation to postoperative complications and survival outcomes. Expanding this aspect would strengthen the study’s clinical relevance and place the findings within the context of modern treatment paradigms.

This is a very interesting remark and I thank the reviewer. In this optic, we are collecting other data in order to evaluate the impact of chemo-immunotherapy (perioperative treatment) vs chemotherapy for all lung resection; actually, the data do not yet have a statistical value and strength. Concerning chemo-immunotherapy and pneumonectomy, the schedule or mandatory intervention of pneumonectomy in the therapeutic decision making, is an exclusion criterion for perioperative treatment. So it is difficult to compare chemo-immunotherapy and chemotherapy for pneumonectomy, as surgical intervention. But this is an insightful and important suggestion for other surgical intervention. Thanks.

Reviewer 3 Report

Comments and Suggestions for Authors

I thank the authors for this interesting paper, which should be published after some clarifications:
1- The patients' cases were discussed at MDT. But how was the decision made for neoadjuvant chemotherapy?
2- The authors should explain in more depth why patients with neoadjuvant chemotherapy had more complications?
3- Results (line 156): Did you perform 405 or 406 pneumonectomies? Is there a typo there?
4- Neoadjuvant treatments to reduce tumor burden play an extremely important role in managing several tumors. The authors should also highlight and discuss the importance interventional treatments, especially chemoembolization but also thermal ablation for managing lung cancer. For example check/cite following studies:
1- Vogl et al. (PMID: 39462003 PMCID: PMC11513032).
2- Xu et al. (PMID: 35371971 PMCID: PMC8965054).

Author Response

REVIEWER 3

I thank the authors for this interesting paper, which should be published after some clarifications:

1-The patients' cases were discussed at MDT. But how was the decision made for neoadjuvant chemotherapy?

Thanks for the important remark. We weekly perform our MDT where we discuss all oncological/surgical patients. The decision-making process process has always been carried out in close collaboration between oncologists and surgeons, precisely to understand how to obtain optimal treatment.

However, since the center is the main national referral center for lung cancer, we have other patients, having started treatment elsewhere and turning to us for surgery.

As debated in the paper, neoadjuvant treatment is usually reserved for locally advanced stage (IIIA stage, N2 lymph node involvement) at the time of diagnosis, potentially resectable, in order to achieve disease downstaging, in terms of size and extension to nearby organs and to reduce the extent of the subsequent surgical resection.

2- The authors should explain in more depth why patients with neoadjuvant chemotherapy had more complications?

I thank the reviewer.

In the discussion section you can find all the molecular, pathological mechanisms and histological changes that are the basis of possible respiratory complications of patients.

“The spectrum of possible chemotherapy-related lung damages is obviously broad, ranging from subclinical modifications to an acute toxicity……….    Referring to the subclinical modifications mentioned above, the most frequent lung damage is at the level of the alveolo-capillary membrane ……..   From a molecular point of view, chemotherapy promotes the presence of inflammatory mediators and cells, such as interleukines or cytotoxic factors at the level of alveolo-capillary membrane…..   This subclinical process impacts on the histological and anatomopathological composition of lung tissue. It has demonstrated that chemotherapy can cause severe lung injury, predisposing patterns of severe and diffuse interstitial involvement, inter-stitial chronic inflammation, fibrosis…….”

 The types of respiratory complications are described both in the table 2, in the results and in the discussion. We did not focus on explaining each single complication to avoid losing the focus of the paper.

3- Results (line 156): Did you perform 405 or 406 pneumonectomies? Is there a typo there?

We excluded only one patient from the analysis because, at the time of the paper, he had not yet reached 120 days of follow-up

4- Neoadjuvant treatments to reduce tumor burden play an extremely important role in managing several tumors. The authors should also highlight and discuss the importance interventional treatments, especially chemoembolization but also thermal ablation for managing lung cancer. For example check/cite following studies:

1- Vogl et al. (PMID: 39462003 PMCID: PMC11513032).

2- Xu et al. (PMID: 35371971 PMCID: PMC8965054).

I really thank the reviewer for the suggestion. Our aim was to evaluate the impact of neoadjuvant treatment for the patient undergoing pneumonectomy, in order to evaluate if preoperative CT is still acceptable in our modern era (chemo-immunotherapy or target therapy). The eminent papers, cited by the reviewers, investigate the possibility of chemoembolization/microwave ablation, for local ablation or for the relapse treatments. I believe, respectfully to the reviewer's opinion, we risk going completely off topic if we get bogged down in the article

Round 2

Reviewer 2 Report

Comments and Suggestions for Authors

Dear Authors,

Thank you for your thoughtful responses and for incorporating previous suggestions regarding the stratification of chemotherapy regimens and the role of immunotherapy in the neoadjuvant setting. Your clarifications provide additional context and strengthen the manuscript. However, there are still some aspects that could be further refined to enhance the clarity and impact of the study.

  1. Interpretation of Statistical Trends
    Some statistical findings, such as the association between neoadjuvant chemotherapy and ARDS (p=0.06), are close to but do not reach significance. While these findings may suggest a trend, it would be helpful to clarify their clinical relevance without overinterpreting the results. A more explicit discussion of potential confounding factors and the study’s power to detect differences in smaller subgroups would improve the robustness of the conclusions.

  2. Clarification on Patient Selection and Preoperative Assessment
    Given the conclusions regarding increased perioperative risk, a more detailed discussion of patient selection criteria would be beneficial. Specifically, while lung function tests such as DLCO and CPET were introduced in 2017, their potential role in identifying high-risk patients could be more explicitly discussed. This would help contextualize how future preoperative assessments might refine surgical decision-making.

  3. Discussion on the Future of Surgical Decision-Making
    Your explanation regarding the exclusion of pneumonectomy from perioperative chemo-immunotherapy trials is valuable. It would be useful to highlight this limitation more explicitly in the discussion to clarify why direct comparisons are not feasible at this stage. Additionally, a brief mention of how emerging treatment paradigms (e.g., targeted therapies or adaptive immunotherapy strategies) may influence pneumonectomy indications in the future would further enhance the study’s relevance.

Overall, the manuscript is well-structured, and the revisions have strengthened its clarity. Addressing these final refinements will further improve the precision of the conclusions and their applicability in clinical practice.

Author Response

REVISIONS

Thank you for your thoughtful responses and for incorporating previous suggestions regarding the stratification of chemotherapy regimens and the role of immunotherapy in the neoadjuvant setting. Your clarifications provide additional context and strengthen the manuscript. However, there are still some aspects that could be further refined to enhance the clarity and impact of the study.

  1. Interpretation of Statistical Trends

Some statistical findings, such as the association between neoadjuvant chemotherapy and ARDS (p=0.06), are close to but do not reach significance. While these findings may suggest a trend, it would be helpful to clarify their clinical relevance without overinterpreting the results. A more explicit discussion of potential confounding factors and the study’s power to detect differences in smaller subgroups would improve the robustness of the conclusions.

Thanks for the remark

We add this limitation into the text, underlying the not reached significativity and the trend observation

“In our analysis the correlation between induction therapy and incidence of postoperative respiratory complications is very clear, both in the simple and adjusted analysis. In particular, the risk of developing simple respiratory complications is double. Concerning ARDS, but it becomes even triple, Even if the statistical significance is not reached in this last subgroup of patients. Indeed we observe a trend of significance that suggests us to understand how the development of ARDS can be strictly linked to preoperative chemotherapy. This means that a pre-treated patient will have a three times greater risk of developing post pneumonectomy ARDS”

  1. Clarification on Patient Selection and Preoperative Assessment

Given the conclusions regarding increased perioperative risk, a more detailed discussion of patient selection criteria would be beneficial. Specifically, while lung function tests such as DLCO and CPET were introduced in 2017, their potential role in identifying high-risk patients could be more explicitly discussed. This would help contextualize how future preoperative assessments might refine surgical decision-making.

Thanks for the remark.

We add some sentences into the discussion section, regarding respiratory functional tests.

“In light of the possible new respiratory scenario, arising after inductive therapy, an “ex novo” deep cardio-respiratory study (spirometry, DLCO, cPET and lung perfusion scan) is essential in view of a subsequent pneumonectomy. In particular, we perform a complete post-induction therapy reassessment, consisting in CPET and lung perfusion scan. This is essential to evaluate the preoperative (cut-off VO2 max > 15 ml/min/kg) and predicted-postoperative (cut-off VO2max > 10 ml/min/kg) VO2max, more than DLCO and simple spirometry, in order to evaluate respiratory post-operative morbidity and mortality. Indeed CPET provides holistic assessment of patient’s physiologic status, depending on several factors: respiratory, cardiovascular, musculoskeletal, circulatory, training and effort”

  1. Discussion on the Future of Surgical Decision-Making

Your explanation regarding the exclusion of pneumonectomy from perioperative chemo-immunotherapy trials is valuable. It would be useful to highlight this limitation more explicitly in the discussion to clarify why direct comparisons are not feasible at this stage. Additionally, a brief mention of how emerging treatment paradigms (e.g., targeted therapies or adaptive immunotherapy strategies) may influence pneumonectomy indications in the future would further enhance the study’s relevance.

Thanks for this other remark.

We add other concepts into the discussion section

“Nevertheless, there are other questions to be asked: what is the price of this advantage in patients undergoing pneumonectomy? Are all patients, considered suitable for surgery before induction therapy, also considered suitable afterwards? Indeed, in this optic, the need for pneumonectomy (based on preoperative staging) is sometimes viewed as a con-traindication to performing neo-adjuvant chemo-immunotherapy.

We must then consider patients with diseases sensitive to targeted therapies (EGFR, ALK), in which the prognosis could be linked to the molecular therapy rather than to the intervention. From this perspective, therefore, neoadjuvant chemotherapy, alone, with all its burden of complications, would lose even more meaning.

From this perspective, it is therefore mandatory to carefully select potentially pneumonectomy eligible patients”

Overall, the manuscript is well-structured, and the revisions have strengthened its clarity. Addressing these final refinements will further improve the precision of the conclusions and their applicability in clinical practice.

Reviewer 3 Report

Comments and Suggestions for Authors

The authors replied sufficiently to most of my comments.

I don’t have further comments.

Author Response

Thanks for your consideration